# Involvement of Expression of miR33-5p and ABCA1 in Human Peripheral Blood Mononuclear Cells in Coronary Artery Disease

**DOI:** 10.3390/ijms25168605

**Published:** 2024-08-07

**Authors:** Yazmín Estela Torres-Paz, Ricardo Gamboa, Giovanny Fuentevilla-Álvarez, Guillermo Cardoso-Saldaña, Rocío Martínez-Alvarado, María Elena Soto, Claudia Huesca-Gómez

**Affiliations:** 1Phisiology Department, Instituto Nacional de Cardiología Ignacio Chávez, Juan Badiano No. 1. Col. Sección XVI, Mexico City 14380, Mexico; yazminestela@hotmail.com (Y.E.T.-P.); rgamboaa_2000@yahoo.com (R.G.); fuentevilla_alvarez@hotmail.com (G.F.-Á.); 2Endocrinology Department, Instituto Nacional de Cardiología Ignacio Chávez, Juan Badiano No. 1. Col. Sección XVI, Mexico City 14380, Mexicoorssino@yahoo.com (R.M.-A.); 3Research Direction, Instituto Nacional de Cardiología Ignacio Chávez, Juan Badiano No. 1. Col. Sección XVI, Mexico City 14380, Mexico; mesoto50@hotmail.com

**Keywords:** miR, ABCA1, monocytes, coronary artery disease

## Abstract

MicroRNAs (miRs) are small non-coding RNAs that regulate gene expression post-transcriptionally and are crucial in lipid metabolism. ATP-binding cassette transporter A1 (ABCA1) is essential for cholesterol efflux from cells to high-density lipoprotein (HDL). Dysregulation of miRs targeting *ABCA1* can affect cholesterol homeostasis and contribute to coronary artery disease (CAD). This study aimed to investigate the expression of miRs targeting *ABCA1* in human monocytes, their role in cholesterol efflux, and their relationship with CAD. We included 50 control and 50 CAD patients. RT-qPCR examined the expression of miR-33a-5p, miR-26a-5p, and miR-144-3p in monocytes. Logistic regression analysis explored the association between these miRs and CAD. HDL’s cholesterol acceptance was analyzed using the J774A.1 cell line. Results showed that miR-26a-5p (*p* = 0.027) and *ABCA1* (*p* = 0.003) expression levels were higher in CAD patients, while miR-33a-5p (*p* < 0.001) levels were lower. Downregulation of miR-33a-5p and upregulation of *ABCA1* were linked to a lower CAD risk. Atorvastatin upregulated *ABCA1* mRNA, and metformin downregulated miR-26a-5p in CAD patients. Decreased cholesterol efflux correlated with higher CAD risk and inversely with miRs in controls. Reduced miR-33a-5p expression and increased *ABCA1* expression are associated with decreased CAD risk. miR deregulation in monocytes may influence atherosclerotic plaque formation by regulating cholesterol efflux. Atorvastatin and metformin could offer protective effects by modulating miR-33a-5p, miR-26a-5p, and *ABCA1*, suggesting potential therapeutic strategies for CAD prognosis and treatment.

## 1. Introduction

Atherosclerosis, an inflammatory disease, is the primary cause of coronary artery disease (CAD) [1] through intima and middle layer thickening in coronary arteries due to monocyte infiltration under the artery lining [2,3,4,5]. Several working groups have tried to identify new mediators capable of regulating the expression of genes involved in the formation of atherosclerosis plaques. miRs are small non-coding single stranded RNAs that can regulate a large amount of mRNA, either by promoting its degradation or inhibiting its translation [6].

In previous studies by our group, we have examined miRs as regulatory agents of atherosclerotic plaque genes [7] and found that human monocytes express inflammatory and oxidative stress-related miRs, whose abnormal expression is associated with an elevated risk for CAD [8]. Although other elements influence cardiovascular diseases, such as in the case of reverse cholesterol transport (RTC), cholesterol is a key player in every stage of atherosclerosis development [8], with one of the miRs involved with lipid metabolism and cardiovascular disease being miR-33-5p. The intronic miR-33a-5p, encoded within SREBP-2, contributes to atherosclerosis development by influencing lipid metabolism and cardiovascular diseases [9,10]. Previous studies in mouse models reveal that miR-33a-5p influences reverse cholesterol transport (RCT) by targeting ATP-binding cassette transporter A1 (ABCA1), responsible for cholesterol efflux to apolipoprotein A-I (apoA-I) and enabling high-density lipoprotein (HDL) production [11,12,13]. miR-144-3p and miR-26a-5p regulate *ABCA1* expression, inhibiting cholesterol efflux to apo-A1 [14,15,16]. The present work aimed to examine miR-33a-5p, miR-144-3p, and miR-26a-5p expression in monocytes, their connection to the target gene *ABCA1*, as well as its role in cholesterol efflux, and their correlation with CAD.

## 2. Results

### 2.1. Characteristics of the Study Population

One hundred subjects were recruited at the National Cardiology Institute, “Ignacio Chávez”, Mexico City; comprising 50 patients diagnosed with CAD and 50 subjects as the control group (CG). Table 1 shows the main biochemical and anthropometric parameters of these groups. The cohort with CAD consisted of 86% men and 14% women. The CAD group showed significantly higher values of age (*p* < 0.001), glucose (*p* < 0.001), systolic blood pressure (SBP) (*p* < 0.001), and diastolic blood pressure (DBP) (*p* < 0.001), and significantly lower values of total cholesterol (TC) (*p* < 0.001) and HDL-C (*p* < 0.001) compared to the CG. Hypertension was present in 64% of CAD patients and diabetes in 41%.

### 2.2. Expression Levels of miR-33a-5p, miR-144-3p, miR-26a-5p, and ABCA1 in the Monocytes of the Subjects

As shown in Figure 1A, the level of miR-33a-5p in monocytes was significantly lower in the CAD group compared with the CG [median: 4.09 (min. 0.30–max. 15.98) vs. median: 9.43 (min. 0.61–max. 50.75); *p* < 0.001, respectively]. miR-144-3p did not present significant differences between both groups (CAD and CG, respectively) [median: 5.61 (min. 0.21–max. 70.00) vs. median: 7.95 (min. 0.41–max. 77.53); *p* = 0.806] (Figure 1B). Conversely, miR-26a-5p was significantly higher in the CAD group compared with the CG [median: 10.63 (min. 0.41–max. 52.65) vs. median: 5.26 (min. 0.23–max. 43.94); *p* = 0.027, respectively] (Figure 1C). *ABCA1* mRNA expression in monocytes was significantly higher in the CAD group compared to the CG (median: 13.82 (min. 5.69–max. 47.08) vs. median: 9.84 (min. 1.44–max. 26.85); *p* = 0.003) (Figure 1D).

### 2.3. Association of Clinical Parameters and miR/ABCA1 Expression with the Risk of Developing CAD

Binary logistic regression analysis was used to identify whether the deregulation of miR expression influences the development of CAD using the clinical parameters and miRs that were shown to be statistically significantly different between the CG and the CAD group (sex, age, total cholesterol, HDL-C, glucose, SBP, DBP, miR-33a-5p, miR-26a-5p, miR-144-3p, and *ABCA1*). Table 2 shows that the decrease in the expression of miR-33a-5p and the increase in *ABCA1* could have a protective effect since they are associated with a lower risk of developing CAD. Meanwhile, HDL-C age and SBP were weakly associated with CAD.

### 2.4. miR/ABCA1 Expression with the Use of Drugs

As most CAD patients were medicated with metformin, atorvastatin, or antihypertensive drugs, a Mann–Whitney U test was performed between the CG and medicated and unmedicated CAD patients to determine whether the expression of the studied miRs/ABCA1 varies according to the use of any of these drugs (Table 3). *ABCA1* mRNA expression significantly increased in the atorvastatin-medicated CAD group compared to the control group (*p* = 0.001) and the non-medicated CAD group with atorvastatin (*p* = 0.011). In contrast, compared to the CG, miR-33a-5p expression maintained a significant decrease in CAD patients with significant differences in both the medicated (*p* < 0.001) and non-medicated (*p* = 0.001) groups with atorvastatin.

In another way, in patients medicated with metformin, miR-26a-5p expression levels were higher in the unmedicated CAD patients compared with the CG (*p* = 0.029), while expression in CAD patients was lower for miR-33a-5p and *ABCA1* mRNA, when compared to the CG, as well as in unmedicated versus medicated CAD patients (*p* ≤ 0.05). Meanwhile, no differences were found in the expression of miRs or *ABCA1* between the medicated and unmedicated groups due to the consumption of antihypertensive drugs.

### 2.5. miR-33a-5p Expression Cut-Off Points

Previous studies have reported an opposite effect on the expression of miR-33a-5p in patients with hypertension, so to evaluate the discriminative ability of miR-33-5p in patients with CAD, an ROC curve was performed. Our results showed that the cut-off point in patients with the disease was higher than the CG (7.6833 and 4.6582, respectively). The cut-off points for the atorvastatin-medicated and non-medicated groups were similar (3.7471 and 3.5633, respectively).

### 2.6. Efflux Cholesterol in CAD

The capacity of HDL in cholesterol efflux from our study subjects was evaluated with the J774 cell line of mouse macrophages to elucidate the capacity of HDL in cholesterol efflux. We found a significant decrease in cholesterol efflux in CAD patients compared to the CG (3.03 ± 0.16 vs. 3.66 ± 0.19, respectively; *p* = 0.013) (Figure 2), and this decrease was associated with an increased risk of developing CAD (Table 4). As we found that patients medicated with statins had a significant increase in the expression of *ABCA1*, which is key in cholesterol transport, we decided to examine whether this also affects cholesterol efflux. We found that not taking statins when having CAD is associated with a decrease in cholesterol efflux and, therefore, an increased risk of disease (Table 5).

### 2.7. Relationship between miRs/ABCA1 and Efflux Cholesterol

To investigate the relationship between miR expression and HDL’s ability to capture cholesterol from macrophages, a Pearson test was conducted comparing the control group (CG) and the CAD group (Table 6). The analysis revealed that the three study miRs negatively correlated with cholesterol efflux, with significant correlations observed only in the CG (*p* < 0.05). For *ABCA1*, a positive and significant correlation with cholesterol efflux was identified (r = 0.321, *p* = 0.030), and this was also the case exclusively in the CG. Given that the findings indicated that atorvastatin may influence cholesterol efflux, CAD patients were categorized based on their usage of this drug. In the non-medicated group, a negative correlation between the study miRNAs and cholesterol efflux was observed, though it did not reach statistical significance. Additionally, no significant associations were found between cholesterol efflux and anthropometric data.

## 3. Discussion

This study identified a decrease in miR-33a-5p expression and an increase in *ABCA1* mRNA expression in the monocytes of CAD patients, both of which are associated with a decrease in the development of this disease.

This finding is relevant because there are reports of overexpressing *ABCA1* in atherogenic mice, which reduces cholesterol accumulation and plaque size [14]. In addition, studies in cell lines and mouse models have indicated that the overexpression of miR-33a-5p and miR-144-3p decreases *ABCA1* expression and plasma HDL-C levels, reducing cholesterol efflux to ApoA-I [17,18]. Other studies have shown that the inhibition of miR-33-5p in mice with atherosclerosis leads to a significant increase in *ABCA1* expression, promoting the reverse transport of cholesterol and the reduction of atherosclerotic injury [12,17,18,19]. The correlation between miR-33-5p expression and *ABCA1* levels, as well as the association of clinical risk factors with CAD, have previously been established.

In our study population, most CAD patients were diagnosed with T2DM and hypertension, leading them to take statins, metformin, and antihypertensives. A possible explanation for why miR33a expression levels are low, contrary to what has been reported in different works, could be the presence of medications [20,21] as the CAD population presented a high prevalence of diabetes mellitus and hypertension. We categorized the CAD patients based on their usage of these medications. According to other groups, within one month of atorvastatin treatment, hypercholesterolemic patients experienced a significant decrease in miRNA-33a-5p expression [22]. In patients treated with atorvastatin, miR-33a-5p levels significantly decreased in both medicated and unmedicated groups. We determined the cut-off points for this miR using ROC curve analysis among the study groups. In the CAD group, the miR-33a-5p cut-off point was higher than in the control group, but no significant differences were observed between the medicated and non-medicated CAD groups. However, we do not rule out the possibility that atorvastatin inhibits the expression of this miR in monocytes from CAD patients, and that, due to this, the decrease in miR-33a-5p in these cells has a cardioprotective effect. The insufficient number of untreated patients might have hindered us from observing significant disparities.

On the other hand, there is evidence that atorvastatin treatment enhances *ABCA1* mRNA expression in macrophage THP-1 cells, boosting cholesterol transfer to ApoA-I [23]. In our results, patients who consumed atorvastatin showed a significant increase in *ABCA1*. According to bioinformatics analysis studies, statins can change the expression of specific miRs in atherosclerosis-related cells and impact Rho GTPase pathway regulation in monocytes and platelets in patients receiving statin therapy for unstable angina [24,25]. Statins inhibit the signaling functions of Rho GTPase proteins by preventing the production of isoprenoids [26]. Statins significantly contribute to cellular function regulation and atherosclerosis pathogenesis [27], and by inhibiting the Rho GTPase pathway, they modulate various miRNAs to confer additional atheroprotective effects. Moreover, inhibition of this pathway also results in increased activation of peroxisome proliferator-activated receptor γ (PPARγ), a transcription factor that, in turn, activates hepatic nuclear receptor X (LRX), which regulates multiple lipid metabolism genes, including *ABCA1* [28]. Therefore, according to our results, atorvastatin could increase *ABCA1* expression in monocytes by inhibiting mevalonate synthesis. In addition to the presence of the drugs, another possible explanation for the low expression of miR33a in CAD could be due to the presence of cardiovascular risk factors per se.

In contrast, atorvastatin did not affect miR-26a-5p expression; however, we found that a significant increase in its expression was only observed in patients who did not consume hypoglycemic agents, specifically metformin. On the other hand, patients who consumed metformin presented expression levels similar to those of the control group. Furthermore, our results showed a negative correlation between *ABCA1* and the expression of miR-26a-5p. However, it should be noted that our study is the first in humans to show that the increase in miR-26a-5p in monocytes could be involved in the formation of atherosclerotic plaques and that, in addition, its expression could be inhibited by treatment with metformin.

Likewise, we assessed the capacity of human subjects’ HDL to promote cholesterol efflux in an experimental setting using mouse macrophages. Our data, consistent with the work of other groups [29,30,31], showed a significant decrease in cholesterol efflux in patients with CAD compared to the control group, which was associated with a higher risk of developing this disease. Furthermore, it was found that atorvastatin treatment reduces the risk of low cholesterol efflux. However, no correlation was found between HDL levels and cholesterol efflux. HDL’s functionality might have decreased, accounting for these findings. A decrease in HDL’s capacity to remove cholesterol from cells leads to its accumulation within macrophages, resulting in foaming and the expansion of atherosclerotic plaques. The size subfractions of HDL determine both the efflux rate and its correlation with CAD. The distinct roles of various HDL subfractions in cellular cholesterol uptake could account for the inconsistent correlation between HDL cholesterol levels and cardiovascular risk assessments [32]. Xian-Ming et al. demonstrated that cholesterol efflux is most efficient with small, dense HDL3b and HDL3c particles. This suggests that strategies aimed at increasing the levels and enhancing the cholesterol efflux capacity of these HDL species in vivo might be more effective in preventing atherosclerosis than those that primarily elevate the levels of larger HDL particles. [33]. The size of HDL is a subject of controversy. Higher *ABCA1* mRNA expression in CAD patients is associated with decreased cholesterol efflux. Macrophage ABCA1 promotes reverse cholesterol transport in vivo. Wang et al. investigated the IL-1 signaling pathway’s impact on type 2 diabetes and atherosclerosis. In vivo, ABCA1 and ABCG1 are the macrophage receptors responsible for reverse cholesterol transport [34]. The study conducted by [21] revealed that THP1 cells, when incubated with [3H]-cholesterol and either atorvastatin or simvastatin, exhibited enhanced cholesterol efflux as a result of *ABCA1* upregulation by the statins. However, it is necessary to increase the sample size of our non-medicated group to verify if the cholesterol efflux is even more reduced in patients who do not consume atorvastatin and, in this way, verify that this drug increases cholesterol efflux.

### Limitations

The study’s sample size of 100 patients may limit the generalizability of the findings. Further research is needed to explore the mechanistic pathways and long-term effects of modulating these miRs with treatments such as atorvastatin and metformin. Another limitation of the work was that it could not match both groups studied by age and sex. However, in our population, it is difficult to find subjects aged over 50 years with no comorbidity. Finally, it would be interesting to study patients with subclinical atherosclerosis and patients with CAD who have had the disease for several years to determine whether miR expression increases or decreases as the disease progresses.

## 4. Materials and Methods

### 4.1. Patient Population

A total of 100 Mexican subjects, 50 patients diagnosed with CAD and 50 subjects without CAD, who were considered as the control group (CG), were recruited at the National Institute of Cardiology, “Ignacio Chávez”. The inclusion criteria for both groups were older than 40 years and agreed to participate by signing informed consent. The criterion for the control group was not presenting with any comorbidity; to ensure that the patients in the control group did not have atheroma or subclinical atherosclerosis, their carotid intima-media thickness (cIMT) was evaluated using ultrasonography. They were recruited from blood bank donors and through brochures posted in Social Services centers. The CAD group, previously reviewed by a group of specialists, presented with angiographically proven obstructive CAD, which is defined as a disease that causes angina symptoms related to stress or exercise due to ≥50% narrowing in the left main artery or ≥70% narrowing in one or more major arteries, and underwent elective primary coronary bypass surgery. Patients were excluded from both groups if they had liver disease, kidney disease, cancer, untreated abnormal functioning of the thyroid gland, infectious processes, corticosteroid treatment, non-Mexican ancestry, and contaminated or insufficient samples. All participants answered standardized and validated questionnaires to obtain information on their family and medical history, alcohol and tobacco consumption, and physical activity.

Diabetes mellitus was defined as glucose values ≥126 mg/dL for at least 3 months. Hypertension was considered when the patient had values ≥130 mmHg in systole and ≥90 mmHg in diastole. Dyslipidemia was defined as abnormal lipid levels; alcoholism was defined according to the Michigan Alcoholism Screening Test (MAST) [34] and the AUDIT trial [35]. Smoking was described as smoking at least one cigarette per day in the previous 6 months.

Fifty patients with CAD were studied, of which 31 (62%) were medicated with atorvastatin. The doses were: 20 mg (23%), 40 mg (33.4%), 60 mg (5.1%), and 80 mg (38.5%). The duration of treatment varied between 3 to 12 weeks at the discretion of the treating physician; while the patients medicated with metformin were 40 (80%).

Additional information was gathered from each subject’s clinical records. The research protocol 18-1075 was approved by the Institute’s Research and Ethics Committee. Informed consent to participate was signed by all patients, considering the ethical principles for medical research with human beings, as stipulated in the Declaration of Helsinki and modified by the Tokyo Congress, Japan (WMA, 2013) [36].

### 4.2. Arterial Intima-Media Measurement

The control group was evaluated for carotid intima-media thickness (cIMT) to ensure they did not have atheroma or subclinical atherosclerosis. The cIMT was assessed by a specialist in high-resolution sonography using high-resolution B-mode SonoSite Micromaxx ultrasound system (SonoSite Inc., Bothell, WA, USA) with a 13–6 MHz transducer, as previously described [7].

### 4.3. Blood and Plasma Samples

Blood samples were obtained by venipuncture after 12 h of fasting in tubes with ethylenediaminetetraacetate (EDTA-Na) for plasma and monocyte isolation. Plasma was separated immediately by centrifugation for determination of the lipid profile, total cholesterol (TC), triglycerides, high-density lipoprotein cholesterol (HDL-C), low-density lipoprotein cholesterol (LDL-C), and glucose.

### 4.4. Laboratory Analysis

Glucose, TC, and triglycerides were analyzed using enzymatic colorimetric methods (Roche-Syntex/Boehringer Mannheim, Mannheim, Germany). HDL-C was measured after precipitation of low-density and very-low-density lipoproteins by phosphotungstate/Mg^2+^ (Roche-Syntex), and LDL-C was estimated by the equation of Friedewald [37], modified by De Long [38]. All assays were under an external quality control scheme (Lipid Standardization Program, Center for Disease Control in Atlanta, GA, USA). We followed the National Cholesterol Education Project (NCEP) Adult Treatment Panel (ATP III) guidelines and thus defined dyslipidemia with the following levels: cholesterol ≥ 200 mg/dL, LDL-C ≥ 130 mg/dL, HDL-C < 40 mg/dL for men and <50 mg/dL for women, and triglycerides ≥ 150 mg/dL.

### 4.5. Isolation of PBMCs and RNA Extraction

This protocol for the isolation of PBMCs and RNA extraction is based on established methods. Whole blood collected in EDTA tubes was diluted 1:1 with 1X PBS—1% heparin and subsequently added to a solution of Ficoll-Histopaque (10771, Sigma-Aldrich, St Louis, MO, USA). The obtained peripheral blood mononuclear cells (PBMCs) were subjected to positive selection to obtain monocytes using CD14 mAb-coated microbeads (Miltenyi Biotec, Bergisch Gladbach, Germany), as previously described [7]. Total RNA, including miRs, was extracted from monocyte samples using Tripure™ isolation reagent (Roche Diagnostics, Indianapolis, IN, USA). Total RNA, at −80 °C, was stored.

### 4.6. miR Reverse Transcription and Quantitative PCR

A reverse transcription reaction (RT-qPCR) was performed in samples of total RNA from monocytes to obtain the cDNAs of the study miRs, using the specific primers for the mature forms of each one through the TaqMan miR RT kit (TaqMan^®^ Advanced miRNA cDNA Synthesis Kit, Applied Biosystem, Foster City, CA, USA, Catalog Number A28007). For miR-33a-5p (hsa-miR-33a-5p), miR-26a-5p (hsa-miR-26a-5p), and miR-144-3p (hsa-miR-144-3p), real-time quantitative reverse transcription polymerase chain reaction (qRT-PCR) was quantified using a commercial kit (TaqMan Gene Expression Assay, Applied Biosystem, Foster City, CA, USA) for miRs, employing the CFX96TM Touch Real-Time PCR Detection System (Bio-Rad, Hercules, CA, USA). Cycling conditions were 2 min at 50 °C and 10 min at 95 °C, followed by 40 cycles of 15 s at 95 °C and 1 min at 60 °C. Expression levels were measured in duplicate and normalized with the reference gene *RNU6B* (NR_002752). The relative expression was calculated with the comparative threshold cycle (CT) method, and data were analyzed using the 2^−ΔΔCt^ method [39].

### 4.7. mRNA Reverse Transcription and Quantitative PCR

RT-qPCR was performed using one µg of the total RNA for cDNA synthesis according to the High-Capacity cDNA Reverse Transcription kit (Applied Biosystem, Foster City, CA, USA). The cDNA was stored at −20 °C. *ABCA1* (Hs01059137_mL) and was measured using a commercially available kit (TaqMan Gene Expression Assay, Applied Biosystem, Foster City, CA, USA) employing the CFX96TM Touch Real-Time PCR Detection System (Bio-Rad, Hercules, CA, USA). The cycling conditions were 2 min at 50 °C and 10 min at 95 °C, followed by 40 cycles of 15 s at 95 °C and 1 min at 60 °C. The gene expression levels were determined in duplicate and normalized with the reference gene *HPRT* (Hs99999909_m1).

### 4.8. Cholesterol Efflux Capacity

J774A.1 mouse macrophages were cultured in RPMI medium 1640 supplemented with 10% FCS. Cells were labeled for 24 h in the presence of an ACAT inhibitor (2 µg/mL CP113818; a gift from Pfizer) using 0.5 mL/well of 1 µCi/mL [1,2-3H] cholesterol (Perkin Elmer) in RPMI plus 1% fetal bovine serum. To upregulate *ABCA1* in J774 cells, we incubated the cells for 16 h with 0.5 mL/well medium containing 0.3 mmol/L Cpt-cAMP (Sigma-Aldrich) and 0.2% bovine serum albumin in RPMI. Values at time zero were obtained from cell wells harvested before the addition of the lipid acceptor. The percentage of 3H-cholesterol released by cells is calculated as (c.p.m. in medium/c.p.m. time zero) × 100. Efflux was the fraction of total cellular cholesterol released in 4 h to isolated HDL lipoprotein (20 µg/mL) added to each well. All samples were performed in triplicate on the same plate, avoiding inter-day variation.

J774A.1 is a cell line isolated from the ascites of an adult female mouse with reticulum cell sarcoma. It can be used in cell biology and immunology research. Some salient features are: it is morphologically the most homogeneous among other macrophage cell lines and possesses surface components peculiar to macrophages, such as Mac-3 antigen, C3b, and Fc receptors, among other features [40].

### 4.9. Statistical Analyses

The data were analyzed in the SPSS version 22 program (SPSS Inc., Chicago, IL, USA). A descriptive analysis of all variables was performed, and the results were expressed as the mean ± standard deviation (SD); for discontinuous variables, the results were represented with the median (min–max). Comparison between groups was conducted using a Student’s *t*-test for continuous variables and chi^2^ for discrete variables. A Mann–Whitney U test was performed for the variables that did not present a normal distribution, as well as to evaluate outcomes of the control group and patients with CAD who were medicated and non-medicated with statins and antihypertensive and hypoglycemic agents. Subsequently, binary logistic regression analysis was conducted to explore the association between study miRs and CAD. ROC curve analysis was performed to determine miR-33a-5p cut-off points. Finally, a correlation analysis was performed using the Pearson test; these were normalized by calculating the logarithm for expression data. Results with *p*-values < 0.05 were considered significant.

## 5. Conclusions

In summary, this study demonstrates that atorvastatin treatment significantly reduces miRNA-33a-5p expression and increases *ABCA1* mRNA expression in CAD patients, which correlates with the primary clinical risk factors of CAD. These changes suggest that atorvastatin may enhance cholesterol efflux and reduce atherosclerotic plaque formation, thereby lowering CAD risk. Additionally, overexpression of *ABCA1* in atherogenic models has been shown to decrease cholesterol accumulation and plaque size, highlighting the therapeutic potential of targeting these molecular pathways. The correlation between miR-33a-5p, *ABCA1*, and clinical risk factors reinforces the significance of these biomarkers in CAD management. Further studies with larger sample sizes are needed to validate these findings and explore the impact of other medications such as metformin on these pathways. Ultimately, atorvastatin and metformin may offer protective benefits in CAD through modulation of miR-33a-5p and *ABCA1*, serving as potential therapeutic targets for predicting and evaluating treatment success.

## Figures and Tables

**Figure 1 ijms-25-08605-f001:**
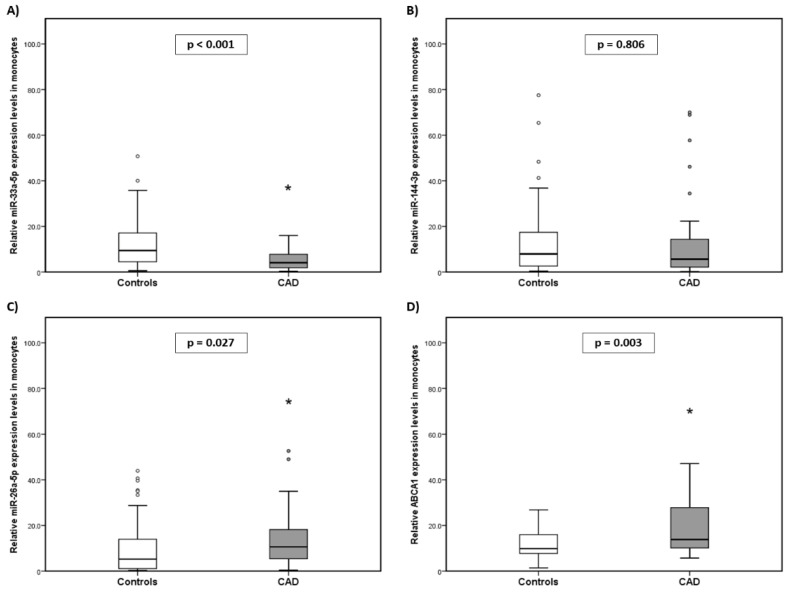
Comparison between HC and CAD patients’ expression levels of the following in monocytes. (**A**) miR-33a-5p, (**B**) miR-144-3p, (**C**) miR-26a-5p, and (**D**) *ABCA1*. The data were normalized to RNU6B. The data are expressed as medians (min.–max.) (Mann–Whitney U test). * *p* > 0.05.

**Figure 2 ijms-25-08605-f002:**
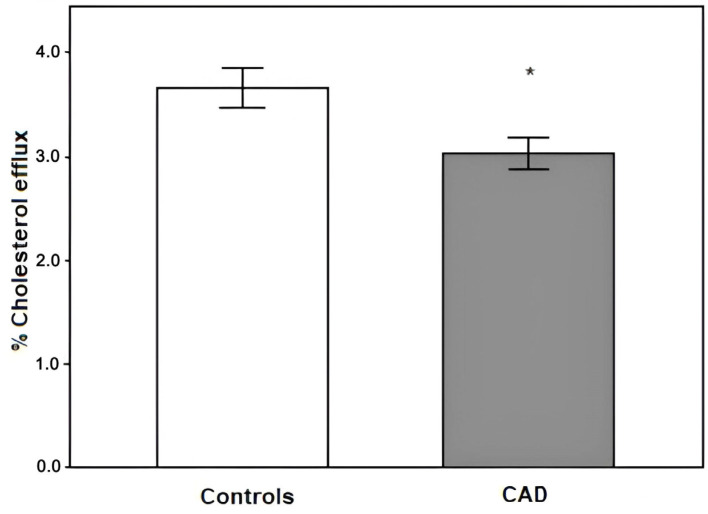
Cholesterol efflux from study groups; * *p* = 0.013. The data are expressed as means ± SE.

**Table 1 ijms-25-08605-t001:** Biochemical and anthropometric parameters of the study population.

Variable	Controls (*n* = 50)	CAD (*n* = 50)	*p*
Age (years)	48.67 ± 6.47	62.45 ± 13.00	<0.001
Sex % M/W	60.7%/39.3%	86.0%/14.0%	0.002
BMI (kg/m^2^)	27.75 ± 3.78	26.54 ± 4.00	0.968
Total cholesterol (mg/dL)	171.06 ± 24.47	134.50 ± 47.05	<0.001
HDL-C (mg/dL)	45.80 ± 14.75	31.07 ± 8.23	<0.001
LDL-C (mg/dL)	102.86 ± 23.17	88.43 ± 51.00	0.069
Triglycerides (mg/dL)	131.48 ± 49.86	140.43 ± 60.77	0.411
Statins %	0%	62.0%	<0.001
Glucose (mg/dL)	93.76 ± 7.82	117.83 ± 43.23	<0.001
Diabetes %	0%	41.0%	<0.001
Hypoglycemic agents %	0%	20.0%	<0.001
SBP (mmHg)	112.43 ± 9.12	126.31 ± 20.76	<0.001
DBP (mmHg)	70.43 ± 6.01	79.59 ± 12.66	<0.001
Hypertension %	0%	64.0%	<0.001
Antihypertensive %	0%	48.0%	<0.001
Smoking %	9.5%	12.0%	0.532
Alcoholism %	0%	2.0%	<0.001
cIMT	0.6026	-	-

The data are expressed as means ± SD (Student’s *t*-test) or percentages (chi^2^ test). BMI: Body mass index, HDL-C: high-density lipoprotein cholesterol, LDL-C: low-density lipoprotein cholesterol, SBP: systolic blood pressure, DBP: diastolic blood pressure, CF: cardiac frequency, cIMT: carotid intima-media thickness.

**Table 2 ijms-25-08605-t002:** Association of clinical parameters and miR/*ABCA1* expression with the risk of developing CAD.

	χ^2^	OR	CI 95%	*p*
Age	6.177	1.088	1.018–1.163	0.013
HDL-C (mg/dL)	8.863	1.180	1.058–1.314	0.003
SBP (mmHg)	5.951	1.071	1.014–1.133	0.015
miR-33a-5p	4.139	0.135	0.020–0.929	0.042
miR-26a-5p	3.149	3.105	0.888–10.853	0.076
*ABCA1*	10.975	0.060	0.011–0.318	0.001

χ^2^: chi^2^ value; OR: odds ratio; CI 95%: confidence interval (binary logistic regression analysis).

**Table 3 ijms-25-08605-t003:** Comparison of miR/ABCA1 expression between the control group and patients with CAD medicated and unmedicated with atorvastatin and metformin.

	**Controls**	**CAD** **Non-Medicated Group with Atorvastatin**	***p*1**	**CAD** **Medicated Group** **with Atorvastatin**	***p*2**	***p*3**
	*n* = 50	*n* = 19		*n* = 31		
miR-33a-5p	9.43 (0.61–50.75)	3.44 (1.20–15.63)	0.001	5.06 (0.31–15.98)	<0.001	0.169
miR-26a-5p	5.26 (0.23–43.95)	11.51 (1.13–19.46)	0.131	8.70 (0.41–52.66)	0.117	0.876
miR-144-3p	7.95 (0.41–77.53)	5.15 (0.21–57.70)	0.354	9.07 (0.50–70.00)	0.557	0.284
*ABCA1*	9.84 (1.44–26.85)	11.26 (6.49–14.36)	0.812	18.35 (6.58–47.08)	0.001	0.011
	**Controls**	**CAD** **non-medicated group with metformin**	***p*1**	**CAD** **medicated group** **with metformin**	***p*2**	***p*3**
	*n* = 50	*n* = 40		*n* = 10		
miR-33a-5p	9.42 (0.61–50.75)	5.04 (0.30–15.98)	0.001	3.90 (0.84–12.60)	0.001	0.767
miR-26a-5p	5.26 (0.23–43.95)	9.75 (1.07–52.66)	0.029	4.91 (1.13–34.96)	0.851	0.092
miR-144-3p	7.95 (0.41–77.53)	5.58 (0.50–68.93)	0.726	6.03 (0.21–70.00)	0.931	0.840
*ABCA1*	14.04 (0.74–78.72)	7.98 (0.91–33.15)	0.036	4.26 (2.63–21.63)	0.050	0.089

The data are expressed as medians (min–max). *p*1 = CAD non-medicated vs. controls; *p*2 = CAD medicated vs. controls; *p*3 = CAD non-medicated vs. CAD medicated (Mann–Whitney U test).

**Table 4 ijms-25-08605-t004:** Relationship between cholesterol efflux and CAD.

	χ^2^	OR	IC 95%	*p*
Cholesterol efflux	5.71	1.52	1.13–2.77	0.017

χ^2^: chi^2^ value; OR: odds ratio; CI 95%: confidence interval (binary logistic regression analysis).

**Table 5 ijms-25-08605-t005:** Relationship between cholesterol efflux and statins.

	χ^2^	OR	IC 95%	*p*
Statins	4.555	2.659	1.083–6.528	0.033

χ^2^: chi^2^ value; OR: odds ratio; CI 95%: confidence interval (binary logistic regression analysis).

**Table 6 ijms-25-08605-t006:** Correlation between miRs/*ABCA1* and cholesterol efflux.

	Controls		CAD		CAD Non-Medicated Group with Atorvastatin		CAD Medicated Group with Atorvastatin	
	r	*p*	r	*p*	r	*p*	R	*p*
miR-33a-5p	−0.357	0.019	0.005	0.976	−0.022	0.947	−0.157	0.444
miR-26a-5p	−0.592	<0.001	−0.047	0.758	−0.190	0.555	0.093	0.643
miR-144-3p	−0.625	<0.001	−0.107	0.505	−0.124	0.701	−0.162	0.471
*ABCA1*	0.316	0.042	−0.008	0.962	0.362	0.248	−0.016	0.941

r: Correlation coefficient; *p* < 0.05 (Pearson’s test).

## Data Availability

Due to confidentiality agreements, the data underlying this study are not publicly available. Access to the data can be requested through c_huesca@yahoo.com following their confidentiality protocols.

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
