# Peer review of "Involvement of Expression of miR33-5p and ABCA1 in Human Peripheral Blood Mononuclear Cells in Coronary Artery Disease"

_ijms, 2024, doi:10.3390/ijms25168605_

Round 1

Reviewer 1 Report

Comments and Suggestions for Authors

In this study the authors investigated the expression of micro RNAs targeting ATP-binding cassette transporter A1in human monocytes, their impact on cholesterol efflux, and their relationship with CAD. This is a study with well described methodology, interesting and innovative results, and considering the increasing incidence of CAD and the urgent need for novel therapeutic strategies it should be taken into consideration for publishing after minor corrections.

1. Check the punctuations in Abstract and Main text.

2. Enhance the figures in the manuscript by coloring the columns of different groups differently and improving the visibility of text inserted into the figures.

3. In the section methodology, add inclusion and exclusion criteria you used for patients involved in the study. Have you mentioned all comorbidities in investigated groups?

4. In the section methodology, briefly describe unmedicated and medicated groups of patients in the section methodology, such as treatment duration, and etc.

5. Have you found some sex differences regarding the influence of observed medications on miR-33a-5p, miR-26a-5p, and miR-144-3p expressions and ABCA1?

6. Add additional informations of origin of mouse cells in the section Methodology.

7. Check the lines 340-342 and remove the information not related to this study

Author Response

In this study the authors investigated the expression of micro RNAs targeting ATP-binding cassette transporter A1in human monocytes, their impact on cholesterol efflux, and their relationship with CAD. This is a study with well described methodology, interesting and innovative results, and considering the increasing incidence of CAD and the urgent need for novel therapeutic strategies it should be taken into consideration for publishing after minor corrections.

  1. Check the punctuations in Abstract and Main text.

Answer: It was done.

  1. Enhance the figures in the manuscript by coloring the columns of different groups differently and improving the visibility of text inserted into the figures.

Answer: It was done.

  1. In the section methodology, add inclusion and exclusion criteria you used for patients involved in the study. Have you mentioned all comorbidities in investigated groups?

Answer: The inclusion criteria for both groups were older than 40 years and agreed to participate by signing informed consent. The criterion for the control group was not presenting with any comorbidity; to ensure that the patients in the control group did not have atheroma or subclinical atherosclerosis, their carotid intima-media thickness (cIMT) was evaluated using ultrasonography, recruited from blood bank donors and through brochures posted in Social Services centers. The CAD group, previously reviewed by a group of specialists, presented with angiographically proven obstructive CAD and underwent elective primary coronary bypass surgery, which is defined as a disease that causes angina symptoms related to stress or exercise. due to ≥50% narrowing in the left main artery or ≥70% narrowing in one or more major arteries. Patients were excluded from both groups if they had liver disease, kidney disease, cancer disease, untreated abnormal functioning of the thyroid gland, infectious processes, corticosteroid treatment, non-Mexican ancestry, and contaminated or insufficient samples. All the participants answered standardized and validated questionnaires to obtain information on their family and medical history, alcohol and tobacco consumption, and physical activity.

  1. In the section methodology, briefly describe unmedicated and medicated groups of patients in the section methodology, such as treatment duration, and etc.

Answer: Fifty patients with CAD were studied, of which 31 (62%) were medicated with atorvastatin. The doses were: 20 mg (23%), 40 mg (33.4%), 60 mg (5.1%) and 80 mg (38.5%). The duration of treatment varied between 3 to 12 weeks, at the discretion of the treating physician; while the patients medicated with metformin were 40 (80%).

  1. Have you found some sex differences regarding the influence of observed medications on miR-33a-5p, miR-26a-5p, and miR-144-3p expressions and ABCA1?

Answer: We did not find significant differences in any of the expressions of the microRNAs analyzed between men and women (data not shown).

  1. Add additional informations of origin of mouse cells in the section Methodology.

Answer: J774A.1 is a cell line isolated from the ascites of an adult, female mouse with reticulum cell sarcoma. It can be used in cell biology and immunology research. Some salient features are: it is morphologically the most homogeneous among other macrophage cell lines; it possesses surface components peculiar to macrophages, such as Mac-3 antigen, C3b, and Fc receptors; among other features. [41]

  1. Check the lines 340-342 and remove the information not related to this study

Answer: It was done.

Reviewer 2 Report

Comments and Suggestions for Authors

The authors present a study that aimed to investigate the expression of miRs and their relationship with CAD. They included 50 control patients and 50 patients with coronary artery disease. The results showed that a downregulation of miR-33a-5p and upregulation of ABCA1 were linked to a lower CAD risk. The authors suggest that atorvastatin and metformin could offer protective effects by modulating miR-33a-5p, miR-26a-5p, and ABCA1, suggesting potential therapeutic strategies for CAD prognosis and treatment. In general, the manuscript is well-written. I would like to suggest a few points that may clarify the methods of the study

1. Could the authors add more information regarding the CAD evaluation? Were the patients diagnosed only through invasive coronary angiography or CT-coronary angiography was used as well? 

2. The authors imply that "The 237 non-invasive diagnosis relied on detecting myocardial ischemia in individuals with ath- 238 eromatous plaques." What were the non-invasive methods used?

3. What was the dose of Atorvastatin used? I believe this may have an important influence on the mRNA expression.

Author Response

We appreciate your observations and comments .

Review 2

The authors present a study that aimed to investigate the expression of miRs and their relationship with CAD. They included 50 control patients and 50 patients with coronary artery disease. The results showed that a downregulation of miR-33a-5p and upregulation of ABCA1 were linked to a lower CAD risk. The authors suggest that atorvastatin and metformin could offer protective effects by modulating miR-33a-5p, miR-26a-5p, and ABCA1, suggesting potential therapeutic strategies for CAD prognosis and treatment. In general, the manuscript is well-written. I would like to suggest a few points that may clarify the methods of the study

Could the authors add more information regarding the CAD evaluation? Were the patients diagnosed only through invasive coronary angiography or CT-coronary angiography was used as well?

Answer:  The CAD group, previously reviewed by a group of specialists, presented with angiographically proven obstructive CAD and underwent elective primary coronary bypass surgery, which is defined as a disease that causes angina symptoms related to stress or exercise. due to ≥50% narrowing in the left main artery or ≥70% narrowing in one or more major arteries. The diagnosis was through invasive coronary angiography and in some cases in patients with a pretest probability of CHD of 50 % or lower was done through computerized tomography (CT) coronary angiogram.

The authors imply that "The non-invasive diagnosis relied on detecting myocardial ischemia in individuals with atheromatous plaques." What were the non-invasive methods used?

Answer: The diagnostic tests including conventional invasive coronary angiography and, in some cases, non-invasive computed tomography (CT) coronary angiography are used in the diagnosis of coronary heart disease (CHD).

  1. What was the dose of Atorvastatin used? I believe this may have an important influence on the mRNA expression.

Answer: Fifty patients with CAD were studied, of which 31 (62%) were medicated with atorvastatin. The doses were: 20 mg (23%), 40 mg (33.4%), 60 mg (5.1%) and 80 mg (38.5%). The duration of treatment varied between 3 to 12 weeks, at the discretion of the treating physician. It was added in the methods section.

Reviewer 3 Report

Comments and Suggestions for Authors

Below, I have attached comments on the manuscript: 

1. I suggest changing the title to a shorter version, possibly omitting the phrase "treatment patients".

2. The introduction is too brief; it does not demonstrate the necessity for the research.

3. The aim of the study should be more detailed.

4. Gene names should be written in italics.

5. Please add a reference (doi: 10.1177/0003319717706616) to the first sentence in the introduction (L35-36).

6. Methods - Please characterize the control group: how were these healthy volunteers selected, and how was it confirmed that they do not have CAD?

7. Methods - How were the plasma samples prepared, and why was plasma chosen over serum for the biochemical parameter studies?

8. Results - The control group does not appear to be properly selected; there are significant differences in age and gender that can substantially affect results, such as lipid parameters and mRNA expression.

9. cIMT - Was it not measured in CAD?

10. The materials and methods section lacks fundamental definitions, for instance, what do the authors mean by diabetes or hypertension?

11. Consequently, a significant portion of the study based on the analysis of CAD vs. CG does not seem to be conducted properly, as age and gender could have a significant impact.

12. Therefore, the conclusions drawn by the authors cannot be substantiated.

13. The authors presented results on statins and efflux, but their description is trivial and lacks justification for their implementation.

14. The authors did not discuss the limitations of the study, of which there are many, making the discussion deeply incomplete.

Author Response

We appreciate your observations and comments .

Review 3

  1. I suggest changing the title to a shorter version, possibly omitting the phrase "treatment patients".

Answer: Thank you for your suggestion. It was done

"Involvement of expression of miR33-5p and ABCA 1 in human peripheral blood mononuclear cells in coronary artery disease"

  1. The introduction is too brief; it does not demonstrate the necessity for the research.

Answer. The introduction was expanded, and a greater explanation of the work was given.

  1. The aim of the study should be more detailed.

Answer. Thank you for the suggestion, it has been modified.

  1. Gene names should be written in italics.

Answer. Thank you for the suggestion, it was done.

  1. Please add a reference (doi: 10.1177/0003319717706616) to the first sentence in the introduction (L35-36).

Answer: Slomka A, Piekus A, Kowalewski M, Pawliszak W, Anisimowicz L, Zekanowska E. Assessment of the procoagulant activity of microparticles and the protein Z system in patients undergoing off-pump coronary artery bypass surgery. Angiology 2018, 69 (4):247-357.

  1. Methods - Please characterize the control group: how were these healthy volunteers selected, and how was it confirmed that they do not have CAD?

Answer. As explained in the methods section (4.1 and 4.2): The criterion for the control group was not presenting with any comorbidity; to ensure that the patients in the control group did not have atheroma or subclinical atherosclerosis, their carotid intima-media thickness (cIMT) was evaluated using ultrasonography, recruited from blood bank donors and through brochures posted in Social Services centers.

In methods 4.2. Arterial intima-media measurement: 

The control group was evaluated for carotid intima-media thickness (cIMT) to ensure it did not present atheroma or subclinical atherosclerosis. The cIMT was assessed by a specialist in high-resolution sonography using high-resolution B-mode ultrasound equipment (Sonosite Micromax) with a 13-6 MHz transducer, as previously described [6].

  1. Methods - How were the plasma samples prepared, and why was plasma chosen over serum for the biochemical parameter studies?

Answer: For the biochemical tests that were performed in our study, both plasma and serum can be used, however, we used plasma, since having all the components, makes it the preferred option for various clinical biochemistry analyses, providing a more detailed understanding of the composition and functionality of blood.

  1. Results - The control group does not appear to be properly selected; there are significant differences in age and gender that can substantially affect results, such as lipid parameters and mRNA expression.

Answer: Thanks for your observation. Indeed, as indicated, there are differences in age and sex between both study groups, which is a limitation of our work. An adjustment was made by binary logistic regression, considering the clinical parameters to identify whether the deregulation of miR expression influences the development of CAD. (point 2.3). Also, it is very difficult to sample older individuals (>50 years) without any comorbidity in our population (Added in the limitations).

  1. cIMT - Was it not measured in CAD?

Answer: It was not performed on patients with CAD, only on controls, since, as mentioned, it was to rule out the presence of atheroma or subclinical atherosclerosis.

In our patients, diagnostic tests including conventional invasive coronary angiography, and, in some cases, non-invasive computed tomography (CT) coronary angiography are used in the diagnosis of coronary heart disease (CHD). This has been added to the methodology.

  1. The materials and methods section lacks fundamental definitions, for instance, what do the authors mean by diabetes or hypertension?

Answer: Are agree, we added some definitions

Diabetes mellitus was defined as glucose values ≥ 126 mg/dL for at least 3 months. Hypertension was considered when the patient had values ≥ 130 mmHg in systole and ≥ 90 mmHg in diastole. Dyslipidemia was defined as abnormal lipid levels; alcoholism was defined according to Michigan Alcoholism Screening Test (MAST) (DeWood MA, et al1980) and AUDIT trial (Jiang K, et al, 2011). Smoking was described as smoking at least one cigarette per day in the previous 6 months.

  1. Consequently, a significant portion of the study based on the analysis of CAD vs. CG does not seem to be conducted properly, as age and gender could have a significant impact.

Answer: As mentioned in point number 8: there are differences in age and sex between both study groups, which is a limitation of our work. An adjustment was made by binary logistic regression, considering the clinical parameters to identify whether the deregulation of miR expression influences the development of CAD. (point 2.3). We added this point in the limitations section.

  1. Therefore, the conclusions drawn by the authors cannot be substantiated.

Thanks for the observation. As mentioned in the previous point, adjustment was made in the analysis for sex and age and we did not find statistically significant differences. Although we agree that studies with a larger population are required to increase statistical power and verify this point. However, also as mentioned in point 8, finding subjects without any comorbidity and over 50 years of age in our population is very complicated. We added this point in the limitations section.

  1. The authors presented results on statins and efflux, but their description is trivial and lacks justification for their implementation.

Answer: We added a sentence to try to justify the análisis.

The capacity of HDL in cholesterol efflux from our study subjects was evaluated with the J774 cell line of mouse macrophages to know the capacity of HDL in cholesterol efflux. We found a significant decrease in cholesterol efflux in CAD patients compared to CG (3.03 ± 0.16 vs 3.66 ± 0.19, respectively; p= 0.013) (Figure 2), and this decrease was associated with an increased risk of developing CAD (Table 4). Due to we found that patients medicated with statins had a significant increase in the expression of ABCA-1, key in cholesterol transport, we decided to examine whether this also affects cholesterol efflux. We found that not taking statins when having CAD is associated with a decrease in cholesterol efflux and, therefore, an increased risk of the disease (Table 5).

  1. The authors did not discuss the limitations of the study, of which there are many, making the discussion deeply incomplete.

Answer: Thanks for the observation. It has been added:

Limitations

The study's sample size of 100 patients may limit the generalizability of the findings. Further research is needed to explore the mechanistic pathways and long-term effects of modulating these miRs with treatments like atorvastatin and metformin. Another limitation of the work was that it could not match both groups studied by age and sex. However, in our population, it is difficult to find subjects with no comorbidity, with ages over 50 years. Finally, it would be interesting to study patients with subclinical atherosclerosis and patients with CAD who have had the disease for several years, to determine whether miR expression increases or decreases as the disease progresses.

Reviewer 4 Report

Comments and Suggestions for Authors

This is a typical case of salamy slicing, the original article being Overexpression of microRNA-21-5p and microRNA-221-5p in Monocytes Increases the Risk of Developing Coronary Artery Disease. Basically, they added 10 cases to the cases group compared to the original study, and used two different markers. There is a lack of relevant citations of the original source. For example in Table 1, the parameters of the cases groups are identifical with the original study (not detected by ithenticate). There are numerous phrases taken word by word from the original source. As it is, this is an obvious case of research misconduct

Comments on the Quality of English Language

Irellevant

Author Response

We appreciate your observations and comments .

Review 4

This is a typical case of salamy slicing, the original article being Overexpression of microRNA-21-5p and microRNA-221-5p in Monocytes Increases the Risk of Developing Coronary Artery Disease. Basically, they added 10 cases to the cases group compared to the original study, and used two different markers. There is a lack of relevant citations of the original source. For example in Table 1, the parameters of the cases groups are identifical with the original study (not detected by ithenticate). There are numerous phrases taken word by Word from the original source. As it is, this is an obvious case of research misconduct.

Answer: Thanks for the observation. As mentioned throughout the article presented, this work is a continuation of a project, for which the same patients were studied. However, it can be seen throughout the article that the objectives and analyses are different. Previously, the study was carried out by measuring other markers (microRNA 21-5p, 221-5p, 155-5p, and NOS3). In the present work, we focused on evaluating the expression of the microRNAs 33a-5p, 144-3p, 26a-5p, and the expression of the ABCA1 transporter. An outstanding point is the measurement of cholesterol efflux mediated by the ABCA-1 transporter and its participation in CAD.

Reviewer 5 Report

Comments and Suggestions for Authors

Strengths:

  • The study provides valuable insights into the role of microRNAs (miRs) in regulating cholesterol homeostasis and their impact on coronary artery disease (CAD).
  • It effectively demonstrates the association between specific miRs and CAD, highlighting potential therapeutic targets such as miR-33a-5p, miR-26a-5p, and ABCA1.

Limitations that should be mentioned:

  • The study's sample size of 100 patients may limit the generalizability of the findings.
  • Further research is needed to explore the mechanistic pathways and long-term effects of modulating these miRs with treatments like atorvastatin and metformin.
Comments on the Quality of English Language

Minor editing of English language required.

Author Response

We appreciate your observations and comments .

Review 5

The study provides valuable insights into the role of microRNAs (miRs) in regulating cholesterol homeostasis and their impact on coronary artery disease (CAD).

It effectively demonstrates the association between specific miRs and CAD, highlighting potential therapeutic targets such as miR-33a-5p, miR-26a-5p, and ABCA1.

Limitations that should be mentioned:

The study's sample size of 100 patients may limit the generalizability of the findings.

Further research is needed to explore the mechanistic pathways and long-term effects of modulating these miRs with treatments like atorvastatin and metformin

Answer: Thanks for the observation. The work limitations are indeed missing, which have been added:

Limitations

The study's sample size of 100 patients may limit the generalizability of the findings. Further research is needed to explore the mechanistic pathways and long-term effects of modulating these miRs with treatments like atorvastatin and metformin. Another limitation of the work was that it could not match both groups studied by age and sex. However, in our population, it is difficult to find subjects who do not have any comorbidity, with ages over 50 years. Finally, it would be interesting to study patients with subclinical atherosclerosis and patients with CAD who have had the disease for several years, to determine whether miR expression increases or decreases as the disease progresses.

Round 2

Reviewer 3 Report

Comments and Suggestions for Authors

The authors have addressed all my questions and concerns.